# Repeated upslope biome shifts in *Saxifraga* during late-Cenozoic climate cooling

Tom Carruthers[1,12], Michelangelo S. Moerland[1,2,12], Jana Ebersbach[3,4], Adrien Favre[5], Ryan A. Folk [6], Julie A. Hawkins[2], Alexandra N. Muellner-Riehl [3,4], Martin Röser[7], Douglas E. Soltis [8,9], Natalia Tkach [7], William J. Baker [1,10,13], Jurriaan M. de Vos [11,13] & Wolf L. Eiserhardt [1,10,13] ✉

Mountains are among the most biodiverse places on Earth, and plant lineages that inhabit them have some of the highest speciation rates ever recorded. Plant diversity within the alpine zone - the elevation above which trees cannot grow—contributes significantly to overall diversity within mountain systems, but the origins of alpine plant diversity are poorly understood. Here, we quantify the processes that generate alpine plant diversity and their changing dynamics through time in *Saxifraga* (Saxifragaceae), an angiosperm genus that occurs predominantly in mountain systems. We present a time-calibrated molecular phylogenetic tree for the genus that is inferred from 329 low-copy nuclear loci and incorporates 73% (407) of known species. We show that upslope biome shifts into the alpine zone are considerably more prevalent than dispersal of alpine specialists between regions, and that the rate of upslope biome shifts increased markedly in the last 5 Myr, a timeframe concordant with a cooling and fluctuating climate that is likely to have increased the extent of the alpine zone. Furthermore, alpine zone specialists have lower speciation rates than generalists that occur inside and outside the alpine zone, and major speciation rate increases within *Saxifraga* significantly pre-date increased rates of upslope biome shifts. Specialisation to the alpine zone is not therefore associated with speciation rate increases. Taken together, this study presents a quantified and broad scale perspective of processes underpinning alpine plant diversity.

Mountain systems contain an exceptionally high proportion of terrestrial biodiversity, and many narrowly endemic species[1–4]. When scaled by area, they are more diverse than lowland regions, including those renowned for high biodiversity such as the Amazon basin[2]. A defining feature of most major mountain systems is the presence of an alpine zone: a high-elevation biome that is climatically unsuitable for tree growth[5]. The alpine zone is characterised by distinctive plant communities adapted to life above the treeline that contribute significantly to overall mountain plant biodiversity[5,6]. Understanding how

and why alpine plant communities have evolved is therefore key to unpicking the drivers of mountain biodiversity as a whole. Until now, a lack of robust species level molecular phylogenies for alpine clades has constrained efforts to address this issue.

Alpine species in any given region can arise in one of three ways[7–10] (Fig. 1a). First, they can originate via adaptation of non-alpine ancestors to the alpine zone (upslope biome shift). This can result from lineages moving into the alpine zone, or the alpine zone increasing in spatial extent and lineages adapting to the changing extrinsic conditions that

**Fig. 1 | Summary of the study system, and different representative species of *Saxifraga*. a** Illustrates the different mechanisms for assembling diversity within the alpine zone. Movement between the alpine zone and Arctic is shown because this study classifies the Arctic and alpine zone as equivalent. **b** Illustrates the distribution of *Saxifraga* within the different regions that are considered in this study. The raw distribution data is available on Zenodo. **c** *S. oppositifolia* (alpine), Italy. **d** *S. cespitosa* (alpine and non-alpine), Svalbard. **e** *S. sedoides* (alpine and non-alpine), Italy. **f** *S. callosa* (alpine and non-alpine), cultivation. **g** *S. squarrosa* (alpine and non-alpine), Italy. **h** *S. hispidula* (alpine and non-alpine), China. **i** *S. smithiana* (alpine and non-alpine), China. **j** *S. stolonifera* (non-alpine), cultivation. **k** *S. caesia* (alpine), Austria. **l** *S. mutata* (non-alpine), Italy. **m** *S. tangutica* (alpine and non-alpine), China. **n** *S. nigroglandulifera* (alpine and non-alpine), China. Photographs by M.S. Moerland (**e**–**g**, **i**–**n**), J. Ebersbach (**d**, **h**) W.J. Baker (**c**).

this brings about. Second, they can originate via the dispersal of pre-adapted lineages from the alpine zone in other regions (mountain hopping). This can be either long-distance dispersal between disjunct areas of the alpine zone, or stepwise dispersal via previously existing habitat corridors. Besides other alpine areas, climatically similar low-land areas at high latitudes (the Arctic) may also serve as source areas from which lineages can disperse. Finally, alpine species can originate from speciation locally within the alpine zone (alpine speciation). It is challenging to unravel the relative importance of these often highly context dependent processes[7–11].

Plants that inhabit the alpine zone experience extreme conditions that may include dramatic temperature fluctuations, intense ultravio-let radiation, short growing seasons, and low-nutrient soils[5,12]. The complex adaptations required to thrive in these conditions, including novel thermoregulatory and photosynthetic pathways, imply that upslope biome shifts are challenging, potentially reducing their prevalence[9]. In general, biome shifts are also thought to be relatively rare in a variety of other settings[13–15]. However, some biomes that also require highly specialised adaptations, such as the Cerrado or Savan-nah biome, are characterised by highly frequent biome shifts[16]. Thus, it cannot be assumed a priori that upslope biome shifts are rare. Addi-tional complexity in the role of biome shifts also relates to direction-ality. For example, a large-scale analysis of European alpine clades highlights that shifts from higher to lower elevations are comparatively frequent compared to the opposite direction[17], whilst an analysis of tropical alpine systems instead highlights that shifts out of the alpine zone are especially rare[8]. These differences may be caused by the different climatic dynamics in tropical compared to temperate regions, with glacial periods in temperate regions potentially causing more dramatic alterations to the ranges of species in the alpine zone[11].

The highly discontinuous distribution of the alpine zone across the surface of the planet, with long distances between different areas of alpine zone, implies that mountain hopping is infrequent. Never-theless, long-distance dispersal has been shown to be more prevalent than biome shifts in other spatially disjointed biomes such as season-ally dry forest[18,19], and even in the alpine zone dispersal has been shown to be prevalent in some contexts. For example, regional-scale analyses[9,20] have highlighted how dispersal into the Qinghai-Tibet Plateau and Himalayas has been more prevalent than alpine speciation in that area throughout the last 40 Myr. Meanwhile, at a broader scale, dispersal can explain the distribution of the "western New World super radiation" in *Lupinus* which has a disjunct distribution between the Rocky Mountains and the Andes[21]. In *Gentiana* and *Rhodiola*, mountain hopping is likely to have occurred at an even broader scale, explaining distribution patterns of alpine taxa that span Asia, Europe, and the Americas[22,23].

Alpine speciation includes ecological speciation driven by the intense habitat heterogeneity of mountain systems. This is likely to be prevalent in *Lupinus* in the Andes[11,24,25], where different species exhibit a range of growth forms indicative of ecological specialisation[24] and increased rates of adaptation have been shown to occur at coding genes[25]. Alternatively, the topographical heterogeneity of mountain systems can lead to highly fragmented habitats, potentially promoting allopatric speciation[26]. Despite these drivers of alpine speciation, the limited spatial extent and extreme conditions of the alpine zone may reduce the available niche space, potentially limiting opportunities for alpine speciation[27]. The dynamism of mountain systems adds further complexity, with climatic and geological changes underpinning dra-matic alterations to the configuration and extent of different habitats through space and time[5,10,28–30]. This could promote speciation through the generation of niches and fragmentation of species ranges, yet also inhibit speciation by preventing stable populations from persisting long enough to become separate species. These conflicting forces are likely to explain the contrasting and context specific conclusions of previous studies of alpine speciation, with some estimating elevated

rates of alpine speciation relative to background speciation rates and directly linking orogeny to these elevated rates[20,24,26,31,32]; others finding no evidence of elevated rates[17]; and others finding evidence for reduced rates in the alpine zone during the climatic fluctuations of the Pleistocene[26].

The interplay of upslope biome shifts, mountain hopping, and alpine speciation in the formation of alpine plant diversity is complex, and a global perspective of the spatial and temporal dynamics of these processes across a widespread and diverse mountain clade and within a comprehensive phylogenetic framework is lacking. Recent advances that facilitate the generation of large molecular sequence datasets, the inference of robust and densely sampled species level phylogenies[25,33,34], and the estimation of macroevolutionary parameters[35–40], mean that this knowledge gap can be addressed.

Here, we present a species-level time-calibrated phylogenetic tree for the angiosperm genus *Saxifraga*, which primarily occurs in mountain systems[33]. This phylogenetic tree is estimated from 329 low copy nuclear loci, incorporates 407 species (73% of the 557 species of *Saxifraga*[41]) and is analysed alongside a database of regional occur-rence and biome preference information with a biogeographical model. This enables three questions to be addressed concerning the role of upslope biome shifts, mountain hopping, and alpine speciation: (1) What is the relative importance of upslope biome shifts and inter-regional mountain hopping for explaining the occurrence of alpine lineages in different regions? (2) Do geological or climatic changes affect the relative importance of these processes through time? (3) Are upslope biome shifts associated with surges in speciation rates? The widespread distribution of *Saxifraga* (Fig. 1b), coupled with its species richness, existence both inside and outside the alpine zone (Fig. 1c–n), and relatively ancient (late Cretaceous/early Paleocene) origin[42], make it an ideal system for investigating processes underpinning the diver-sity of the alpine zone at broad spatial and temporal scales.

Overall, we show that upslope biome shifts have occurred at a higher rate than inter-regional mountain hopping in *Saxifraga*. This pattern is especially pronounced in the last 5 Myr, implying that global climatic cooling and temperature fluctuations during this period have led to increased pressure and/or opportunity to adapt to the alpine zone. Further, high speciation rates within *Saxifraga* are not associated with adaptation to the alpine zone.

## Results

### Phylogenetic inference

The final species tree estimated in ASTRAL[43] contains 580 tips, 491 of which belong to *Saxifraga*. These represent a total of 407 species, 73% of the species diversity of the genus (Supplementary Fig. 1)[41]. The phylogenetic tree is broadly congruent with previous trees based on ribosomal (ITS) and plastid DNA[33,42]. In particular, major clades delimited in[33] are recovered with high support, although there are some topological changes within and among these clades (Fig. 2a; Supplementary Fig. 1). There are high levels of gene-tree-species-tree conflict, with the most congruent gene trees sharing approximately 30% of nodes with the inferred species tree topology. Gene-tree-species-tree conflict is particularly pronounced among some recently diverged lineages (Supplementary Fig. 1).

### Divergence time estimation

Analyses using *treePL*[44] resulted in a crown node age estimate for *Saxifraga* of ~67 Ma, close to the Cretaceous-Palaeogene boundary (Fig. 2a) and consistent with previous results based on few nuclear and plastid loci[42]. This estimate was also consistent across time-calibrated phylogenies estimated under a range of different assumptions, except when no maximum constraints were used at internal nodes, where the crown node age estimate is in the mid-Cretaceous (Supplementary Table 1, Supplementary Fig. 2). Most major clades within *Saxifraga* are estimated to have originated throughout the Miocene (Fig. 2a),

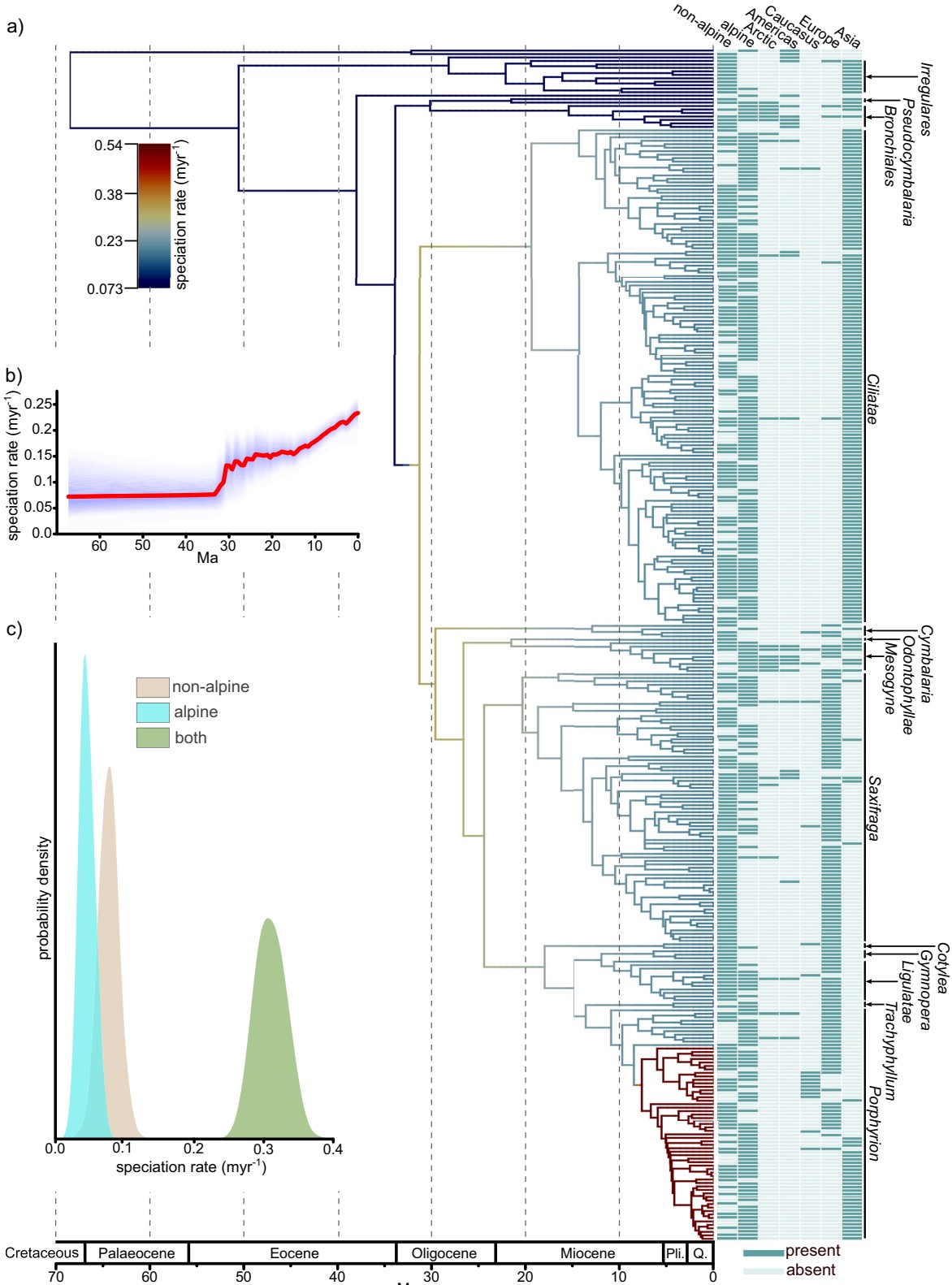

**Fig. 2 | Phylogeny and speciation dynamics in *Saxifraga*. a** Estimated time-calibrated phylogeny, with branch colours showing lineage-specific speciation rates estimated in BAMM, and the matrix referring to the biomes and regions inhabited by each species. Pli. Pliocene, Q. Quaternary. **b** Mean speciation rate through time estimated in BAMM. Red line is the posterior mean estimate, blue shaded area is the posterior distribution, darker shades correspond to a higher posterior probability than lighter shades. **c** Posterior distribution for the speciation rate in each of the three biome categories from the ClaSSE analysis. Raw sequence reads are deposited in the Sequence Read Archive, distribution data are available on Zenodo, the biome and regional assignment database is available in Supplementary Data File 3.

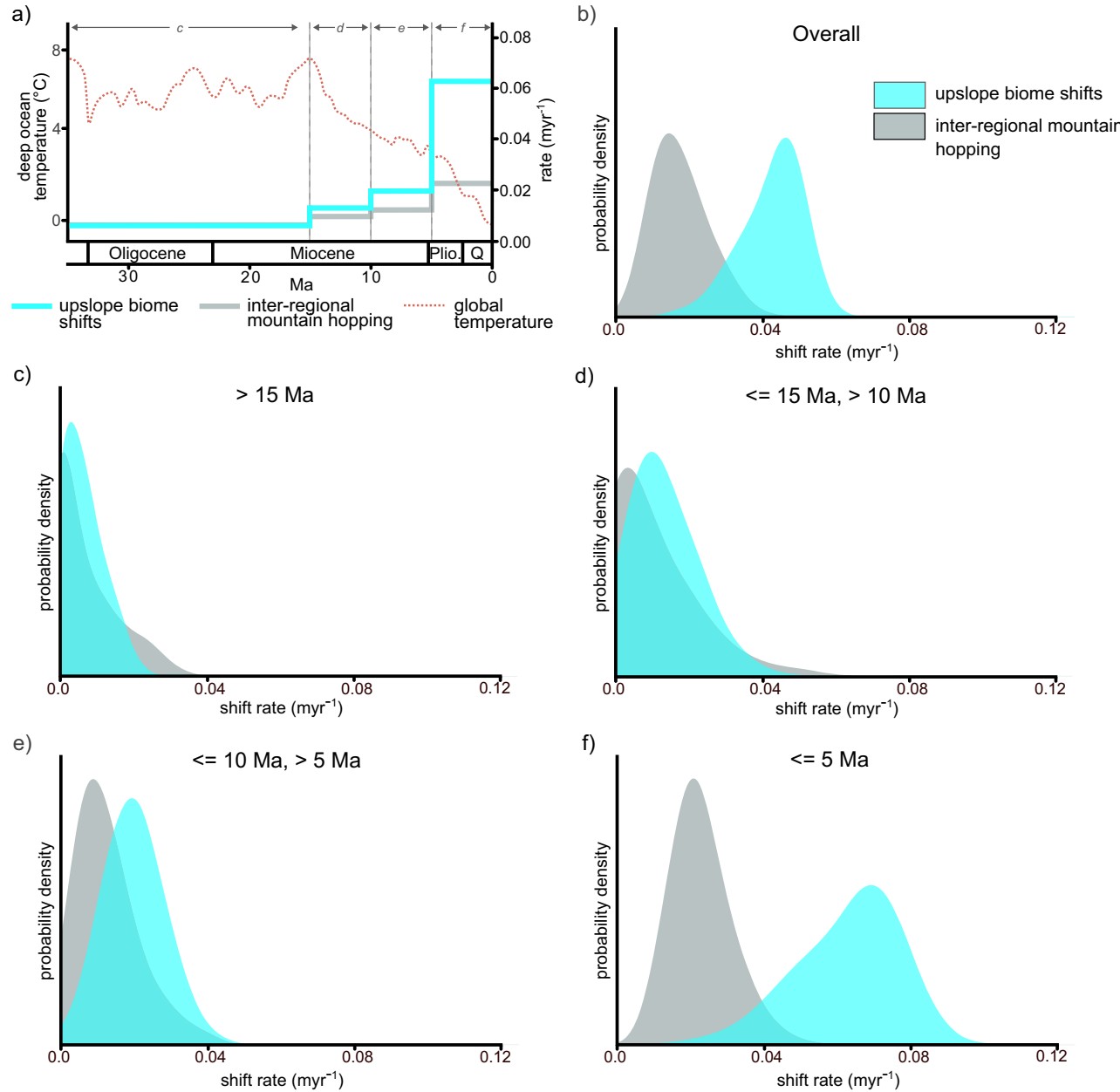

**Fig. 3 | Rates of upslope biome shifts and inter-regional mountain hopping.**
**a** Posterior mean rate of upslope biome shifts and inter-regional mountain hopping through time, as well as global climate through time (estimated by deep sea oxygen isotope records)[52]. The vertical dotted lines delimit the time intervals that are referenced in (**c**–**f**). Pli. Pliocene, Q. Quaternary. **b**–**f** Posterior distributions of rates of upslope biome shifts and inter-regional mountain hopping from the ClaSSE model, with (**b**) showing rates across the entire time-calibrated phylogeny, (**c**) showing rates at times over 15 Ma, (**d**) showing rates 15 Ma or less, but over 10 Ma, (**e**) showing rates 10 Ma or less, but over 5 Ma, and (**f**) showing rates 5 Ma or less. Distribution data are available on Zenodo, the biome and regional assignment database is available in Supplementary Data File 3.

although infrageneric divergence time estimates vary somewhat between methods (Supplementary Table 1, Supplementary Figs. 2, 3).

## Speciation rate estimation
Estimation of lineage-specific speciation rates with BAMM[39] supports two distinct speciation rate increases (Fig. 2a, b): one ~30 Ma at the origin of the clade containing sections *Porphyrion*, *Saxifraga*, and *Ciliatae*, and a further nested speciation rate increase within section *Porphyrion* ~8 Ma. This configuration of rate shifts (or very similar configurations) receives the highest posterior support, although several alternative configurations are recovered with considerably lower support (many of which are very similar to the best configuration) (Supplementary Fig. 4). Estimation of biome specific speciation rates

with the ClaSSE model highlighted that generalists (those inhabiting both the alpine zone and non-alpine biomes) have the highest speciation rate. The rate for such lineages is around four-fold higher than for alpine specialists (corresponding to alpine speciation rates) and species that do not occur in the alpine zone (Fig. 2c).

## Rates of upslope biome shifts and inter-regional mountain hopping
The estimated rate of upslope biome shifts is higher than that of inter-regional mountain hopping (Fig. 3). This pattern is especially pronounced during the most recent 5 Myr (Fig. 3a, f). Notably, the increased rate of upslope biome shifts is temporally concordant with falling global temperatures in the Pliocene, and temperature

fluctuations in the Pleistocene (Fig. 3a). By contrast, major orogenic events within the distribution range of *Saxifraga* substantially predate the timing of increased rates of upslope biome shifts.

The increased rate of upslope biome shifts in the most recent 5 Myr (Fig. 3a, f) also significantly post-dates speciation rate increases estimated in BAMM (Fig. 2a, b). This finding, alongside the result from the ClaSSE analysis where the speciation rate for alpine specialists is lower than that of generalists (Fig. 2c), indicates that specialisation to the alpine zone is not associated with speciation rate increases.

Topological uncertainty in the species tree could potentially bias estimates of upslope biome shifts and inter-regional mountain hopping and their changing rates through time. An incorrect topology could disrupt the association of clades with certain regions or biomes, and thus lead to overestimation of rates of dispersal or biome shifts. Although it is not computationally feasible to account for this comprehensively, for example by repeating the ClaSSE analysis over a distribution of species tree topologies, we highlight that the species tree topology is well supported, and that there is no decrease in the level of support (indicating an increase in topological error) during the last 5 Myr (Supplementary Fig. 5). The increased rate of upslope biome shifts in recent times is therefore unlikely to result from species tree estimation error.

## Discussion

Overall, our results suggest that alpine species in different regions are primarily derived from non-alpine lineages via upslope biome shifts, a process that has intensified substantially within the last 5 Myr. This conclusion is based on the overall estimated rate of upslope biome shifts being ~3 times higher than that of inter-regional mountain hopping (Fig. 3b), and the estimated rate of upslope biome shifts increasing markedly within the last 5 Myr (Fig. 3a, f). The higher rate of upslope biome shifts is especially notable given our analytical framework is biased toward inter-regional mountain hopping, with lineages being able to transition between five regions but only two biome categories (Fig. 1). The recent increase in upslope biome shifts is temporally concordant with, and thus potentially driven by global climatic cooling and increased climatic instability in the Pliocene and Pleistocene (Fig. 3a). Conversely, it is temporally disconnected from, and thus unlikely to be driven by major orogenic events within the distribution range of *Saxifraga*.

We also show that specialisation to the alpine zone is not associated with higher speciation rates: the ClaSSE analysis estimates the highest speciation rates for biome generalists (Fig. 2c), whilst significant speciation rate increases estimated in BAMM substantially predate increased rates of upslope biome shifts (Figs. 2a, b, 3). Rapid alpine speciation does not therefore appear to be a major driver of *Saxifraga* diversity within the alpine zone. Further studies are necessary to determine whether the patterns presented here are replicated in other clades, and the processes that drive potential similarities and differences that exist between clades.

### Upslope biome shifts are more important than inter-regional mountain hopping

Biome shifts and dispersal are fundamentally different mechanisms for assembling regional biotas and because of this their relative importance has been debated extensively[45–47]. Recently, there has been a greater emphasis on dispersal and the closely associated pattern of phylogenetic niche conservatism[13–15] in which dispersing lineages maintain similar habitat preferences regardless of their geographical location.

However, this study illustrates the importance of upslope biome shifts rather than dispersal of pre-adapted lineages from other regions in *Saxifraga* (Fig. 3). Limited dispersal within *Saxifraga* is intuitive given the lack of specialisations for dispersal within the genus[48], and the fact that many sections are near endemic to a particular region

(Fig. 2a). Further, previous analyses have highlighted that long-distance dispersal in *Saxifraga* is rare[42], whilst regional scale analyses from other groups have demonstrated the importance of upslope biome shifts[7,8,10]. This study builds on this work by quantifying the relative rate of upslope biome shifts and dispersal at a broader spatial scale and within a robust phylogenomic framework.

The emphasis on upslope biome shifts does not preclude more prevalent dispersal at narrower spatial scales, as has been demonstrated in several settings including dispersal within and between the Himalayas and Qinghai–Tibet-Plateau and within European mountain systems[9,20,49]. Even within *Saxifraga*, dispersal by wind or large herbivores can occur[50,51]. However, at the broad spatial scale of this analysis, upslope biome shifts are far more prevalent.

### Late-Cenozoic climate cooling as a driver of upslope biome shifts

The increased rate of upslope biome shifts in the last 5 Myr (Fig. 3a, f) is temporally concordant with a continuous fall in global temperatures that began with the mid-Miocene cooling[52], as well as with climate oscillations during the Pleistocene (Fig. 3a). By contrast, it significantly postdates major orogenic episodes within the distribution range of *Saxifraga*, such as the orogeny of the Qinghai-Tibet Plateau around 45 Ma[23]. Climatic change rather than orogeny is therefore more likely to have driven the increased rate of upslope biome shifts. Such a scenario can be explained by falling global temperatures increasing the extent of the alpine zone and therefore providing more opportunities for specialisation to the alpine zone[15]. By contrast, earlier phases of orogeny, when the Earth's climate was far warmer (Fig. 3a), would have likely given rise to less extensive and more fragmented areas of alpine zone. Alongside this, climate oscillations during the Pleistocene may have further increased the rate of adaptation to the alpine zone through the "species-pump effect", where climatic oscillations cause pulses of speciation followed by range expansion into other habitats[53].

### Upslope biome shifts do not precipitate surges in speciation rates

The finding that there is no association between upslope biome shifts and higher speciation rates (Fig. 2), and that upslope biome shifts are instead associated with speciation rate decreases (given that alpine specialists have lower speciation rates than biome generalists and species that do not occur in the alpine zone, Fig. 2c), contrasts with prominent examples of rapid diversification in high elevation and alpine habitats[11,54], such as the explosive diversification of *Lupinus* in the Andes[11,21,24,25]. The differences between this example and our study are likely underpinned by four factors. (1) *Lupinus* in the Andes is far younger than *Saxifraga*, meaning there is a shorter timeframe over which different processes can exert a complex legacy on diversification dynamics. (2) Unlike this study, discussion of *Lupinus* is not focussed specifically on the alpine zone, but more generally on high elevation areas that inevitably provide more opportunities for diversification than solely the alpine zone. (3) Unlike *Saxifraga*, *Lupinus* exhibits significant ecomorphological differentiation[11,21,24,25], implying active adaptive radiation *within* high elevation habitats. Specialisation to the alpine zone in *Saxifraga* by contrast appears to limit the number of niches into which the genus can evolve and therefore diversify. (4) Unlike *Saxifraga*, *Lupinus* has radiated in aseasonal environments. This may be the key factor that enabled more significant ecomorphological adaptation compared to *Saxifraga* species, many of which are distributed at high latitudes and thus subject to the evolutionary constraints of very short growing seasons. Nonetheless, this dissociation between upslope biome shifts and increased speciation rates is consistent with several studies[17,26,27], including a recent analysis of the entire of Saxifragales[55], where increased speciation rates pre-date increased rates of niche and phenotypic evolution. Given our study is undertaken at a very different phylogenetic and ecological scale

compared to ref. 55, the consistency between these studies suggests a disassociation between adaptation and speciation may not be an uncommon phenomenon in plant macroevolution.

### Analysing different scales and developing explanatory hypotheses

This study demonstrates the importance of upslope biome shifts for generating alpine plant diversity across broad spatial scales within *Saxifraga*. Future analyses that integrate findings across different spatial scales may be important, as demonstrated by studies that illustrate prevalent mountain hopping at narrower spatial scales (such as between the Qinghai-Tibet Plateau, Himalayas, and Hengduan Mountains or within the European Alps), and notable differences between regions in how alpine floras were assembled[8–10,17,20,25,27,42]. The comparison of different ecological scales may also be insightful. For example, the low stature of *Saxifraga* may buffer it from the climatic stressors that play a key role in defining the alpine zone[56], highlighting how processes other than adaptive shifts between broadly defined biomes play an important role in the macroevolution of the genus. Alongside this, further investigation of habitats outside the alpine zone with similar characteristics to the alpine zone is necessary, as these habitats may serve as a stepping stone for adaptive shifts into the alpine zone. Nonetheless, the species we classify as alpine zone specialists do not, by definition, occur in such areas. Our definition of upslope biome shifts is therefore likely to capture lineages undergoing broad adaptive changes that specifically enable inhabitation of the alpine zone.

### Concluding comment

Using phylogenomic data for a major mountain plant clade of the Northern Hemisphere, we illustrate the recency with which alpine species have evolved from non-alpine ancestors. Specifically, we suggest that Pliocene and Pleistocene climatic cooling caused a surge in upslope biome shifting as alpine habitats expanded repeatedly. Alpine habitats are now contracting due to rapid ongoing anthropogenic climate warming, and there is evidence that this is already leading to widespread extinction of alpine plant diversity, a pattern that will likely continue in the future[57,58]. However, even in the unlikely event that extinction rates remain constant in alpine habitats, our study suggests that the recruitment of new lineages from lowland floras will wane as alpine habitats shrink[59]. Therefore, alpine plant diversity is likely to be lost if anthropogenic climate change continues unabated.

## Methods

### Taxon sampling

DNA samples were collected from wild populations, living collections, and herbaria, with the aim being to maximise the number of *Saxifraga* species that were sampled. Collection efforts followed a species list based on Plants of the World Online[41]. DNA samples taken from wild and living collections were desiccated in silica gel. However, most samples were sourced from historic collections from nineteen herbaria, with this material also being stored in silica gel prior to DNA extraction. The recorded collection dates of these vouchers ranged up to 186 years before DNA extraction. Herbarium voucher information is given in Supplementary Data File 1. Additional raw sequence data were included from ref. 55 (Supplementary Data File 2), to include a broad sampling of Saxifragales. Taxon selection outside *Saxifraga* was based on the percentage of overlapping loci that were available for representatives of relevant clades for setting node age constraints.

### Probe design for target capture

**Locus selection.** High-throughput sequencing through targeted enrichment[60,61] was used because the method is highly efficient with fragmented DNA from herbarium collections[62,63]. Biotinylated oligonucleotide baits (probes) were redesigned for loci previously selected

as putative low-copy genes throughout the order Saxifragales[55]. The *Saxifraga* DNA sequence data resulting from[55] showed that probes designed to work universally among Saxifragales had unequal retrieval rates among *Saxifraga* s.s. More specific probes may therefore produce a higher target capture efficiency within the genus. In addition, higher capture efficiency is warranted due to the strong reliance on herbarium-sourced DNA samples. To supplement the pool of exon regions to select from, reference exon regions that were previously selected for targeted enrichment probes for the closely related genus *Micranthes* (Saxifragaceae)[34] were also included. Among the targeted exons, the 301 loci from the 'Saxifragales targets' and the 295 loci from the '*Micranthes* targets' shared 101 loci, through an initial screening with the 'Map to reference' function in Geneious v8.0.5 (www.geneious.com). The remaining 495 unique loci were further used to assess capture efficiency and orthology within *Saxifraga*.

To evaluate the performance of all loci from the Saxifragales and *Micranthes* probes in *Saxifraga*, pre-generated targeted enrichment sequencing results from target-capture sequencing by[55] were used. Sequence divergence within *Saxifraga* was accounted for through the incorporation of target-capture reads from six species, representing different sections: *S. fortunei* Hook. (sect. *Irregulares*), *S. magellanica* Poir. (sect. *Saxifraga*), *S. parnassifolia* D.Don (sect. *Ciliatae*), *S. pulchra* Engl. & Irmsch. (sect. *Porphyrion*), *S. rebunshirensis* Sipliv. (sect. *Bronchiales*), and *S. rotundifolia* L. (sect. *Cotylea*). HybPiper v1.3.1[64] was then used to align the trimmed reads to the 495 target loci, using the standard options for mapping through BWA[65] and assembly through SPAdes v3.11[66].

HybPiper was used to return only exon regions and flag potential paralogous copies. Potential paralogous copies were identified for 36 loci and were visually assessed in a phylogeny with all taxa and their multiple copies of the locus. Trees were estimated through sequence alignment with default settings in MAFFT v7 (--auto)[67] and phylogenetic inference in FastTree v2.1.7 (using the options -nt -gtr)[68]. Tree topologies did not suggest ancient duplication within *Saxifraga* as paralogs within taxa formed clades. Hence, no additional measures were taken to select which copies to use, and instead the putatively orthologous copy selected through HybPiper was used.

Finally, loci were selected where target-capture data retrieved DNA sequences of sufficient length for one or more *Saxifraga* representatives, with those sequences then prepared for probe design. Exon sequences were aligned to the target loci with MAFFT (--auto). To guarantee overlap among sequences for a locus, sequences with a length of <200 bases were omitted. For each locus, all remaining sequences of *Saxifraga* spp. were kept, including sequences of *S. stolonifera* Curtis already present in the target sequences of ref. 55. Loci were omitted entirely if no *Saxifraga* spp. representatives remained. If fewer than four *Saxifraga* spp. were available for a locus a maximum of four non-*Saxifraga* target sequences from ref. 55 and ref. 34 was included. Priority was given to taxa that were most closely related to *Saxifraga* and within the Saxifragaceae alliance accepted by ref. 69. The taxa included are *Itea virginica* L. (Saxifragaceae), *Micranthes caroliniana* (A.Gray) Small (Saxifragaceae), *Mitella pentandra* Hook. (Saxifragaceae), and *Ribes* aff. *giraldii* Janczewski. (Grossulariaceae).

**Bait design.** After testing for orthology and overlap among existing sets of probes, custom probes for 329 loci were designed, of which 281 were part of the universal probes for the order Saxifragales and 48 were unique to the probes targeting *Micranthes*. The length of exon regions that the reference sequences covered ranged from 318 to 4017 bp, with a total length of 1,663,233 bp for all target regions combined. Custom myBaits biotinylated baits were designed and produced by Arbor Biosciences (formerly Mycroarray; Ann Arbor, Michigan, USA). A total of 1463 unique exon sequences were used by Arbor Biosciences for bait design to produce 38,731 probes with 2x tiling.

## Sequence data generation

**DNA extraction.** Samples were ground in 2 ml tubes with two stainless steel beads in an MM400 mixer mill (Retsch Inc.). For living collections and plants collected from the wild, DNA was extracted with the DNeasy Plant Mini Kit (Qiagen, Manchester, UK), using the manufacturer's protocol with at least 1 h of initial incubation time. For herbarium specimens, an adapted CTAB extraction method[70] was used, followed by an extra cleaning step of the total DNA with QIAamp DNA Mini Kit spin columns (Qiagen, Manchester, UK). DNA fragment size estimation was performed on 1% agarose gel and where needed fragments were sheared through sonication with a Covaris ME220 Focused-ultrasonicator™ (Covaris Inc., Woburn, Massachusetts, USA). Sample shearing was implemented to produce fragment lengths of approximately 300 or 600 bp, depending on available fragment length distribution of total DNA and available sequencing platforms. Sonication was not performed on total DNA where the gel image indicated most fragments were already at or below a 300 bp threshold. The DNA extraction method per sample is given in Supplementary Data File 1.

**Library preparation.** DNA library preparation was performed with the NEBNext® Ultra™ II Library Prep Kit and NEBNext® Multiplex Oligos for Illumina® (Dual Index Primers set 1 and 2) (New England BioLabs, Ipswich, MA, USA). Total DNA concentrations were measured with a Quantus™ Fluorometer (Promega Corporation, Madison, WI, USA), and size-selection and cleaning steps performed with Agencourt AMPure XP Beads (Beckman Coulter, Indianapolis, IN, USA). An initial quantity of 200 ng of total DNA was used as specified in the protocol, or less if no more was available after extraction procedures (down to 11 ng). Size selected libraries were cleaned to 300 bp with Agencourt AMPure XP Bead Clean-up (Beckman Coulter, Indianapolis, IN, USA), while only a cleaning step was performed for samples with highly degraded (peak of fragment length <200 bp) and low total DNA (<50 ng). A set of 96 'high-quality' libraries with long mean total library length was size-selected at 500–600 bp. Libraries were amplified through 8–12 PCR cycles, followed by bead clean-up. Library fragment length, DNA concentration and the presence of primer dimers were assessed with a 4200 TapeStation System (Agilent Technologies, Santa Clara, CA, USA). If library concentration was <10 ng/μl, an additional 4-6 PCR cycles were performed followed by bead clean-up.

Library pools were enriched with the custom myBaits kit following the myBaits User Manual v3.02 with the hybridization step set to 65 °C for ~20 h. Target amplification was then performed through 8–12 PCR cycles with KAPA HiFi 2X HotStart ReadyMix PCR solution (Roche, Basel, Switzerland). Quality control for each pool was then performed through use of a TapeStation, after which pools were normalized to 4 nM, multiplexed, and denatured as preparation for sequencing on an Illumina MiSeq machine (Illumina, San Diego, CA, USA) at the Jodrell Laboratory at the Royal Botanic Gardens, Kew (Richmond, UK).

**Sequencing.** Sequencing of most enriched libraries was performed with four 2 × 150 bp v2 Illumina® MiSeq reagent kits. The 96 'high-quality' libraries containing longer insert lengths were pooled together and were sequenced with both a 2 × 250 bp v2 and a 2 × 300 bp v3 Illumina® MiSeq reagent kit. Additionally, library preparation, enrichment, and sequencing of 84 DNA extractions was undertaken by Arbor Biosciences (Ann Arbor, Michigan, USA) using their workflow and an Illumina NovaSeq PE150 platform. All raw read files generated in this study are deposited in the National Centre for Biotechnology Information (NCBI) Sequence Read Archive under BioProject PRJNA672815 and PRJNA923645.

## Sequence assembly and phylogenetic inference

Adapters and low-quality bases were removed using Trimmomatic v0.36[71], employing recommended settings for Illumina® paired-end reads (ILLUMINACLIP:TruSeq3-PE.fa:2:30:10 LEADING:3 TRAILING:3 SLIDINGWINDOW:4:15 MINLEN:36). Before and after trimming sequences were visually assessed using FastQC v0.11.7 [Andrews 2010].

Exon retrieval was performed using HybPiper v.1.3.1[64]. Therein, the trimmed reads were mapped to target DNA sequences through the BWA mapper[65] and assembled through SPAdes v3.11[66]. If multiple distinct contigs were mapped against a reference, all but the longest contig was discarded unless multiple contigs covered over 85% of the target region. After this step the true ortholog for further analysis is selected based on having a mean read depth per base ten times higher than the other contigs. If this is not the case the contig with the highest percent identity to the reference is selected.

Data cleaning steps were performed before and after alignment due to substantial variation in retrieved exon sequence lengths within loci, potentially causing alignment mismatch and long branch attraction. First, any exon sequences were removed that were fewer than 300 bases in length. Then, sequences were removed when significantly shorter than the other available sequences for their respective locus, which was set to 40 percent of median sequence length to reduce the number of short sequences while not discarding too much data. After initial cleaning steps, sequences were aligned in MAFFT (--retree --maxiterate 200)[67]. All sites for which more than 95% of the taxa had no information were removed through Phyutility v2.7.1[72]. Maximum likelihood (ML) gene trees were then estimated with RAxML v8.2 (-f a --# 1 -m GTRGAMMA)[73] and analysed in TreeShrink v1.3.3[74] using centroid re-rooting. Problematic samples, identified as those having a false positive error rate below 0.05, were removed. Sequences were then realigned in MAFFT and gene trees re-estimated in RAxML as described above. From these gene trees a final species tree topology was estimated in ASTRAL v5.6.3[43].

## Divergence time estimation

Six minimum node age constraints were used. These were derived from a literature search of fossil representatives of close relatives of *Saxifraga* (Supplementary Table 2). Two maximum node age constraints were used, one of 125 Ma at the root node (roughly corresponding to the emergence of diverse Eudicot pollen in the Cretaceous[75]), and one of 79 Ma at the *Ribes* stem node (corresponding to the age of the minimum age constraint at the *Ribes* stem node plus two times the largest temporal gap in the *Ribes* fossil record[76]). This maximum constraint is designed to account for gappy-ness in the *Ribes* fossil record. Additional analyses were performed with either no minimum or no maximum constraints at internal nodes to assess the sensitivity of divergence time estimates to these constraints (Supplementary Tables 1, 2).

Gene tree species tree conflict can cause error in molecular branch length estimates (i.e., estimates of the number of substitutions per site on a particular branch) in the species tree, and downstream error in divergence time estimates[77]. Two different analyses were performed that attempt to account for this. First, a branch-wise method for estimating molecular branch lengths in the species tree[77] was used. The method requires individual gene trees that possess molecular branch lengths for individual loci. It then cycles through each branch in each gene tree. Where a branch in the gene tree is topologically congruent with a branch in the species tree (i.e., the same clades are descended from the ancestral and descendant nodes of the branches in the gene tree and species tree) the length of the branch in the gene tree is stored for the relevant branch in the species tree. Once every branch in every gene tree has been cycled through, branch lengths in the species tree are calculated as the mean across all the branch lengths that are stored for the branch across all the gene trees. This method can therefore estimate parameters for the species tree based on the parts of the gene trees that are topologically congruent with the species tree, whilst excluding parts that are not congruent.

Alternatively, a gene shopping approach was used that focuses on loci with topologically congruent gene trees. This method ranks the

loci according to the extent to which they are topologically congruent with the species tree. Loci were then selected according to this ranking until every taxon was represented by at least two loci. This equated to 137 loci. These loci were concatenated and used to estimate molecular branch lengths in RAxML with a GTRGAMMA model, -f e setting to optimise branch lengths and model parameters, and the topology constrained to that estimated in ASTRAL.

Divergence times were estimated in treePL[44]. This requires an input phylogenetic tree with molecular branch lengths, and a set of minimum and maximum node age constraints. Several different input phylogenetic trees and sets of fossil calibrations were used, as described above. Cross-validation was used to select the optimal smoothing value (the parameter in treePL that describes among-branch-substitution-rate-variation). However, alternative smoothing values were also used to quantify the effects of different assumptions about among-branch-substitution-rate-variation. A summary of all estimated time-calibrated phylogenetic trees is shown in Supplementary Table 1.

Following divergence time estimation, subsequent analyses were primarily based on a time-calibrated phylogenetic tree designated as main (Supplementary Table 1). This tree is considered to incorporate the most accurate set of divergence time estimates for several reasons. First, it uses the branch-wise method that specifically samples parts of the gene trees that are topologically congruent with the species tree, unlike gene shopping that incorporates parts of the dataset that are topologically incongruent with the species tree. Second, the implemented fossil calibrations have been assigned following careful evaluation of the literature, and careful consideration of theoretical issues relating to the difference between fossil ages and clade ages. Third, cross-validation has been shown to be relatively effective at estimating the extent of substitution rate variation[27]. However, rates of upslope biome shifts, mountain hopping, alpine speciation, and lineage specific speciation were also estimated in the time-calibrated phylogeny that differs most significantly from main, no maximum (Supplementary Figs. 7–9, Supplementary Note 1). The sensitivity of key biological conclusions to alternative divergence time estimates could therefore be determined.

## Occurrence data
Georeferenced occurrences were downloaded from the Global Biodiversity Information Facility (GBIF) on 30th June 2021[78]. These were cleaned using functions from the CoordinateCleaner R package[79] that remove specimens if they are georeferenced to capital cities, the equator, geographic centroids of countries or regions, GBIF headquarters, or biodiversity institutions; or if they are duplicated, have equal latitude and longitude values, or zero latitude or longitude values. This dataset was augmented with herbarium specimens from Royal Botanic Gardens, Kew (K). In some cases, incorporation of specimens from K used specimens that were not georeferenced. Instead, specimen labels were interpreted in order to assign specimens to a region.

## Definition of regions and biomes
For computational reasons, the ClaSSE analysis (see below) was only feasible for a maximum of five geographic regions. Five regions were defined that reflect historical dispersal barriers, climate, and current discontinuities in the distribution of *Saxifraga*[42]. Following Ebersbach et al.[42] the Northern Hemisphere was divided into three main continental areas: North America, Asia, and Europe. Unlike Ebersbach et al.[42] this study does not have a specific focus on biogeographic connections among Asian mountain ranges. Asia was therefore treated as a single region. However, the Caucasus, situated between Europe and Asia, was treated as a separate region. This area hosts 22 species of *Saxifraga* that are separated geographically from their congeners in Europe and Asia by >1000 km. This separation is far larger than any

spatial discontinuity in any other region. South America, with only two species of *Saxifraga* that likely originated from North America[42], was not recognised as a separate region but merged with North America. Similarly, the African continent is inhabited by very few species, including just one unique species, and was omitted from the study. Because the Arctic, being ecologically highly similar to the alpine zone, can potentially serve as an important stepping stone for dispersal in alpine taxa[80], it was included as a region. The following five regions were therefore included: Americas, Asia, Caucasus, Europe, Arctic.

Within each region, two biomes were delimited: alpine (defined as areas at an elevation above which trees do not grow, thus including the sometimes-distinguished nival zone) and non-alpine. However, in the Arctic region only the "alpine" biome was recognised due to the ecological similarity of Arctic lowlands with lower-latitude alpine habitat. In practical terms, this was achieved by not allowing "Arctic non-alpine" taxa to occur in the subsequent analyses. Taken together, this relatively coarse categorisation of biomes and regions is appropriate for the purposes of this study in that it provides a basis for investigating the processes that underpin diversity in the alpine zone at broad geographical scales.

## Assignment of species to regions and biomes
Specimens were assigned to one of the five regions based on their occurrence data, or, for some of the additional specimens from K, based on interpretation of the specimen label. Regional assignments for species based on georeferenced specimens were then also manually checked following extensive consultation of the relevant literature (Supplementary Data File 3). Meanwhile, biome assignments for species were performed manually based on the expertise of the authors of this study, following extensive consultation of the relevant literature (Supplementary Data File 3), and with reference to the same occurrence data as for the regional assignments. Each species was designated as alpine, non-alpine, or both. Importantly, when assigning species, the primary focus was the dominant biome preference of a given taxon. Therefore, for species which were primarily biome specialists, but occasionally occur in an alternative biome, these species were classified as specialists (alpine or non-alpine) rather than being placed in the "both" category. When assigning species to biomes, decisions were communicated and discussed extensively amongst the authorship. This reduced the risk that assignments were biased by potential differences in how the authors conceptualise different biomes. Although imperfect, this overall approach was the most appropriate given there is an insufficient number of georeferenced specimens with sufficiently accurate and precise coordinates to reliably assign species to biomes based on such coordinates. The final dataset of geographical and biome assignments contained 344 species, of which 97, 106, and 141 were classified as alpine, non-alpine, and both respectively (Supplementary Data File 3).

## Estimating geographical shifts, biome shifts, and speciation using a ClaSSE model
Prior to performing ClaSSE analyses, the time-calibrated phylogeny estimated above was pruned such that it only incorporated the 344 species with biome and regional assignments. Cladogenetic and anagenetic events included in the CLaSSE model are shown in Supplementary Fig. 6. The model incorporates separate speciation, extinction, and anagenetic rates for alpine, non-alpine, and generalist lineages. Sympatric, subset-sympatric, and vicariant speciation events are incorporated. The ClaSSE model was implemented in RevBayes v1.1.2[40].

Rates of upslope biome shifts were based on the number of inferred transitions from biome generalists to alpine zone specialists, whilst rates of inter-regional mountain hopping were based on the number of times alpine specialists immigrated to a new region. To obtain rates, these values were divided by the time interval under consideration (either the time interval incorporated by the

entire dated tree, or more specific time intervals as indicated in Fig. 3).

### Lineage-specific diversification rate estimation

Lineage-specific speciation and extinction rates were estimated in BAMM v.2.5.0[39]. These analyses were performed with the same pruned phylogeny as was used with the CLaSSE model. Appropriate priors were designated with the *setBAMMpriors* function in R from the package BAMMtools[81], the prior number of rate shifts was set to 1, and a global sampling fraction of 0.62 was specified. The analysis was run for 10 million generations, sampling every 50 generations. 1 million generations were discarded as burnin.

### Reporting summary

Further information on research design is available in the Nature Portfolio Reporting Summary linked to this article.

## Data availability

The sequence data generated in this study have been deposited in the Sequence Read Archive (BioProject PRJNA672815 and PRJNA923645). Voucher information for all samples in the molecular dataset are available in Supplementary Data File 1 and Supplementary Data File 2. Alignments and gene trees and geographical distribution data are available on Zenodo at: https://zenodo.org/records/10530358. The final database of biome and regional assignments is available in Supplementary Data File 3. Source data for Fig. 1 is the distribution data on Zenodo. Source data for Fig. 2 is the molecular data deposited in the SRA, distribution data deposited in Zenodo, and final biome and regional assignment database in Supplementary Data File 3. Source data for Fig. 3 is the distribution data deposited in Zenodo, and final biome and regional assignment database in Supplementary Data File 3.

## Code availability

Scripts used to perform analyses and plot results are available at: https://github.com/pebgroup/alpine_saxifraga[82].

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

## Acknowledgements

This work was mostly funded by a grant by the David and Claudia Harding Foundation (Harding Alpine Programme) to the Royal Botanic Gardens, Kew. T.C.'s postdoctoral position was also funded by the Aarhus University Research Foundation (grant AUFF-E-2017-7-19 to W.L.E.) and grants from the Calleva Foundation to the Plant and Fungal Trees of Life (PAFTOL) Project at the Royal Botanic Gardens, Kew. J.M.dV. acknowledges Swiss National Science Foundation grant 310030_185251. R.A.F. acknowledges National Science Foundation (USA) grant NSF DBI-1523667. A.N.M.-R. acknowledges German Science Foundation grant MU 2934/3-1. We thank Robyn Cowan, László Csiba, Penny Malakasi, Juan Viruel, Sidonie Bellot and Grace Brewer for aid with laboratory protocols. In addition, we thank Matthew Jeffery, Andreas Tribsch, Yang Yang and Li Zhang for their help organising and attending field collections. We are also indebted to staff at the many herbaria (B, BASBG, CDBI, E, FIAF, HAL, HUH, JE, KUN, LE, LJU, LY, LZ, M, MB, NEU, W) and living collections (Cambridge University Botanic Garden; Chelsea Physic Garden; Royal Botanic Garden, Edinburgh; Royal Botanic Gardens, Kew; Botanic Garden of Johannes Gutenberg University Mainz; Utrecht University Botanic Gardens) from which we sourced DNA samples.

## Author contributions

Conceptualization: T.C., M.S.M., W.J.B., J.M.d.V., W.L.E.; Methodology: T.C., M.S.M.; Formal analysis: T.C. and M.S.M.; Investigation: T.C. and M.S.M.; Writing—Original Draft: T.C. and M.S.M.; Writing—review & editing: J.E., A.F., R.A.F., J.A.H., A.N.M.-R., M.R., D.E.S., N.T., W.J.B., J.M.d.V., and W.L.E.; Supervision: J.A.H., W.J.B., J.M.d.V., and W.L.E.; Funding acquisition: W.J.B., J.M.d.V., and W.L.E.; Resources: J.E., A.F., R.A.F., A.N.M.-R., M.R., D.E.S., and N.T.

## Competing interests

The authors declare no competing interests.

## Additional information

¹Royal Botanic Gardens, Kew, Richmond, Surrey TW9 3AE, UK. ²School of Biological Sciences, University of Reading, Whiteknights, Reading, Berkshire RG6 6EX, UK. ³Department of Molecular Evolution and Plant Systematics & Herbarium (LZ), Institute of Biology, Leipzig University, D-04103 Leipzig, Germany. ⁴German Centre for Integrative Biodiversity Research (iDiv) Halle-Jena-Leipzig, D-04103 Leipzig, Germany. ⁵Regional Nature Park of the Trient Valley, la Place 24, 1922 Salvan, Switzerland. ⁶Department of Biological Sciences, Mississippi State University, Starkville, MS 39762, USA. ⁷Martin Luther University Halle-Wittenberg, Institute of Biology, Geobotany and Botanical Garden, Dept. of Systematic Botany, Neuwerk 21, 06108 Halle, Germany. ⁸Florida Museum of Natural History, University of Florida, Gainesville, FL 32611, USA. ⁹Department of Biology, University of Florida, Gainesville, FL 32611, USA. ¹⁰Department of Biology, Aarhus University, 8000 Aarhus C, Denmark. ¹¹Department of Environmental Sciences—Botany, University of Basel, Schönbeinstrasse 6, 4056 Basel, Switzerland. ¹²These authors contributed equally: Tom Carruthers, Michelangelo S. Moerland. ¹³These authors jointly supervised this work: William J. Baker, Jurriaan M. de Vos, Wolf L. Eiserhardt. ✉e-mail: wolf.eiserhardt@bio.au.dk

