## [Peer Review File · Nature Communications]

Repeated upslope biome shifts in *Saxifraga* during late-Cenozoic climate coolingReviewers' Comments:

Reviewer #1:

Remarks to the Author:

This study, submitted by Carruthers et al., is an attempt to understand the origins of alpine plant diversity and their changing dynamics through time using the widespread alpine genus *Saxifraga* as an example. Specifically, this study examined the upslope biome shifts, mountain hopping, and alpine speciation processes on alpine plant diversity. The results found that upslope biome shifts are more common than mountain hopping throughout the evolutionary history of *Saxifraga*. Meanwhile, this study found that alpine specialists have lower speciation rates than generalists. I really like this study and am very interested in the scientific questions and study system presented in the current study. However, I have several concerns about the method part.

First, how the current biogeographical framework is suitable for answering the proposed scientific questions on alpine biome assembly needs to be more specific. Here, the author actually modeled the speciation, colonization, and upslope processes among regions for alpine lineages of *Saxifraga* but not the process among real mountain systems (as shown in Figure 1a). For example, Asia includes subtropical mountain regions to cool temperate regions and even islands. This is definitely not a single mountain system, both presently and geologically. The definition of regions may bias the present results. This needs more explanation and details.

Second, this study found a lower speciation rate of alpine lineages and upslope biome shift are more prevalent than dispersal between regions, which is partially contrary to a previous study by Ding et al. (2020). Ding et al. (2020) found that within the Tibet-Himalaya and Hengduan region (THH), alpine lineages in Hengduan Mountain and Himalaya have a higher in situ speciation than local recruitment (similar to upslope biome shift in this study) and colonization (similar to mountain hopping in this study). While the assembly of alpine lineages in the central Qinghai-Tibet plateau is dominated by colonization processes mainly from Hengduan Mountain. Their results suggested that the origins of alpine plant diversity and their changing dynamics through time could be different in different mountain systems. In the current study, the author classified 5 biogeographical regions but did not distinguish the potential differences of the underlying processes among these regions. For example, Asia and Europe contain over 90% of total species of *Saxifraga*. And Asia contains more alpine species but Europe contains more non-alpine species. The underlying process for the assembly of alpine plants of *Saxifraga* could be different. For Arctic America Caucasus, which contains fewer species. I suggested to highlight more on the differences between the present study and Ding et al. (2020) and also add more details. For example, the present study used the genomic data while Ding et al (2020) used sequences of small DNA fragments that were reported by other groups. In addition, the species distributions are and the defined biomes of two studies could be compared.

Third, this study only focuses on one taxonomic group; therefore, the conclusion is probably taxonomically specific. This caveat could be mentioned in

Other minor comments: There are no figure indexes and legends for supplementary figures.

Reviewer #2:

Remarks to the Author:

Review of the manuscript NCOMMS-23-21107 entitled "Repeated upslope biome shifting during late-Cenozoic climate cooling in a diverse alpine plant clade" for Nature Communications

The study investigates patterns in alpine flora assembly using a time-calibrated phylogeny of *Saxifraga*, a flowering plant clade distributed almost globally in mountain systems. Three potential processes are investigated, upslope biome shifts, mountain hopping, and in situ alpine speciation. The relative importance of these processes is analysed by categorizing species into five geographic regions and two biome categories (alpine and non-alpine), and by comparing 'specialists' (alpine or non-alpine) and 'generalists' (species occurring in both biome categories). Estimated are (1) the rates of

shifts among biome categories and (2) speciation rates per biome category. Shift rates are compared for 'specialists' and 'generalists' in total and in four time categories (5, 5–10, 10–15, and >15 Ma). Results indicate that (i) upslope biome shifts are ~3 times higher than that of mountain hopping and that these shift frequencies are accelerated in the last 5 Ma, and (ii) generalists show higher speciation rates than specialists (low 'alpine speciation'). Additionally, lineage-specific speciation rates are estimated, and results indicate that (iii) periods of speciation rate acceleration (at ca. 30 and 8 Ma) in *Saxifraga* predate biome shift acceleration (≤ 5 Ma). The results are discussed in the context of Late-Cenozoic climate cooling, which likely correlates to a spatial extension of alpine areas, as well as Pliocene climate fluctuations. Both are discussed as drivers of upslope biome shifts and these shifts are suggested as the process that primarily generates alpine plant diversity.

From the results presented it seems valid to conclude that "alpine floras of individual mountain systems are primarily derived from non-alpine floras via upslope biome shifts" (line 262f). A flaw here is that the authors do not explicitly state that "mountain hopping" is analysed only between the five global geographic regions and not within "mountain systems," which could, or certainly would, increase the rate of alpine-to-alpine mountain dispersals. This makes the 'mountain hopping' results only valid for global geographic scales, which should be mentioned. What is less clear, however, is whether this "upslope biome shift" (mainly montane to alpine?) will at all entail ecological adaptation. The alpine biome is defined in reference to trees (that do not grow there) and therefore has ecological implications primarily for trees. Whether this holds also for herbs is not investigated in the study and seems at least questionable. For example, a rocky cliff habitat some hundred meters below the tree line might have quite similar abiotic conditions to a similar habitat in the alpine zone (see e.g., Körner & Hiltbrunner 2021 Diversity 13, 383). With this in mind, the statement "that a decoupling between adaptation and speciation may be prevalent in plant macroevolution" (line 337f) is not supported by the evidence presented.

The study is the first that investigates alpine flora assembly on a global scale in a common phylogenetic framework and so is of great significance for alpine ecology, biogeography, macroevolution, and conservation.

The paper is well-written and a pleasure to read. The analytical approach is elegant and the quality of the data seems to be very solid (with the exception of how phylogenetic uncertainty is considered; s. paragraph below). What is less clearly presented is the "Biogeographical reconstruction" (SI line 19) that relies on the ClaSSE model illustrated in Figure S8. A short paragraph (in the SI?) summarizing how mountain hopping and upslope biome shifts are actually obtained from the ClaSSE estimates would be appropriate (also, how was the model set up, e.g., from what kind of distribution did you draw transition rates and diversification rates? et cetera). Also, while assessing submitted data, I was slightly surprised that no time-calibrated tree (the 'key' framework), no 'easy-to-access' data set of regional and biome preferences, and no analytical scripts (RevBayes, etc.) are provided. This 'missing' data does not seem essential for my report but would be great for the scientific community if provided upon publication.

The analytical approach to tree inference seems well done. Phylogenetic uncertainty with respect to gene tree estimation error of branch length is accounted for in the divergence time estimation by considering 'branch-wise' mean length estimates in the input tree, and by the 'gene shopping' approach (although, more sophisticated approaches that also account for topological conflict do meanwhile exist; e.g., Zhang & Mirarab, 2022, <https://doi.org/10.1093/molbev/msac215>). This is, however, not the case for phylogenetic uncertainty with respect to species relationships ('topological conflict' in my understanding) that is, according to the authors, "significant" (e.g., SI line 189; branch length uncertainty and topological- 'relationship of lineages' uncertainty are two different parts of phylogenetic uncertainty, aren't they?). Unluckily, both approaches to account for branch length uncertainty do explicitly ignore the topological information that is in conflict with the inferred species tree. It is unclear to me how phylogenetic uncertainty when fully accounted for would influence results. Uncertainty in divergence time estimates is, as much as possible, convincingly accounted for (see Figs S3, S4, S6, S7). But how does topological uncertainty influence estimated shift rates is neither investigated (certainly, there are computational limits) nor discussed. A statement discussing this potential source of rate estimate biases should be appropriate.

The ClaSSE model is a valid approach and accounts for unsampled or extinct diversity in analyses of

biogeographic range evolution.

The manuscript presents very clearly structured arguments in the introduction and discussion and the reasoning is easy to follow throughout. Besides, sometimes a little more understatement would strengthen the narrative and provide more confidence in the interpretation of results and the conclusions (see, e.g., second paragraph of this review).

Appropriate literature is referred to in the manuscript.

Check ms throughout (also figures) for consistent usage of Ma/Myr.

Hope these comments are helpful. Very much looking forward to seeing this study published!

Kind regards, Nicolai

Reviewer #3:

Remarks to the Author:

General comments: This is a rare study with a high sampling percentage and in general a well conducted study of an amazing group that deserves to get published in Nature Communications! The text could improve by some careful rewording. I included some line-by-line comments. The term alpine and high elevation environments are unfortunately not always used very stringently in literature, and it is easy to compare apples with pears. I would recommend setting this dilemma out in the introduction and read the published literature carefully regarding the definition used and apply it stringently.

I appreciate that the authors seem to have worked extensively on the Qinghai-Tibet Plateau however a more balanced approach to all regions in which *Saxifraga* occurs might be useful when trying to present this as a world-wide model of alpine evolution.

Line 78 this paragraph described well studied processes that do not exclusively occur in alpine habitats but are poorly referenced here with an unfitting citation of a paper looking at orogeny and climatic influence in Tibet-Himalaya-Hengduan mountains. Please reconsider more appropriate citations of the rich scientific literature on the topic!

Line 86: "colonizing" is a very distinctive term in plant biology in addition to social connotation that might be unwanted. Colonizing is probably best reserved for the process of establishing in a previously uninhabited area. As such "colonizing" would not be wrong but I would recommend using immigration instead.

Line 89 while indeed "The relative importance of these processes is challenging to unravel and highly context- and scale-dependent." There is work available that should be at least cited here such as Gehrke 2018 Staying cool: Preadaptation to temperate climates required for colonising tropical alpine-like environments. DOI:10.3897/phytokeys.96.13353

Line 91 I would recommend looking at the literature on tree line more closely as the main factor is not "temperature fluctuations, intense ultraviolet radiation, short growing seasons, and low nutrient soils". Treeline seem to be mainly driven by average temperature and especially average soil temperature! Körner & Paulsen 2004 A world-wide study of high-altitude treeline temperatures. <https://doi.org/10.1111/j.1365-2699.2003.01043.x> And I would recommend including this fact as not to confuse characterised with causing in reader unfamiliar with the subject. I believe the authors wish to express here that plants growing in alpine environments often experience "temperature fluctuations, intense ultraviolet radiation, short growing seasons, and low nutrient soils".

Line 105 both citations point toward work done on Tibetan plant lineages, I find it difficult to accept that as relevant enough to make such a broad general statement here. Please add relevant citations from other parts of the world or remove. This is especially true as the cited work from Favre does not prove that dispersal happened from alpine environments. The publication focuses on geography not climate reconstructions and includes both alpine and subalpine as a single biome.

Line 107 I would recommend to rewrite this as the sentence is confusing (“implies mountain hopping may also be infrequent”?) and there are surely more elegant ways to present this.

Line 110 I find work on island biogeography much more relevant here. There are some beautiful studies such as Alsos et al. 2007 Frequent Long-Distance Plant Colonization in the Changing Arctic. DOI: 10.1126/science.1139178

Line 112 I would move citation 19 to before the full stop and instead add Gehrke & Linder 2014 Species richness, endemism and species composition in the tropical Afroalpine flora DOI:10.1007/s00035-014-0132-0 as a regional study

Line 126 it might be useful to incorporate here the results of Brochmann, et al. History and evolution of the afroalpine flora: in the footsteps of Olov Hedberg. Alp Botany 132, 65–87 (2022). <https://doi.org/10.1007/s00035-021-00256-9>

Line 134 not all of the tree free Andean habitats are alpine habitats, and I would recommend not to use citation 12 here as that paper does not proof this point. I would even doubt that 27 is really proving “alpine speciation rates being elevated relative to background rates”. However, this is not easy and could also be addressed by more careful wording.

Line 145 Please add citation.

Line 149 “mountain hopping” sound repetitive and directional maybe long-distance dispersal might be better used? No matter what the authors decide a definition would be great, it this only between larger geographical regions or how is long-distance dispersal/mountain hopping identified? Fig.2 would indicate that the authors only considered changes on continental scale?

Line 153. I’m not a native English speaker but precipitate has a negative, foreboding connotation for me. You might want to check this.

Line 158 It is a bit unusual to have a paragraph with conclusions at the end of the introduction, but I leave that one to journal style guides.

Line 161 pressure or maybe opportunity (well my glass I half full 😊)

Line 163 see comment on line 153 but since the authors see pressure rather than opportunity maybe increased species rate is indeed negative?

Figure 1: Beautiful! It might be ignorant to new changes but is *Saxifraga hederifolia* from Ethiopia no longer accepted?

Line 190 “Baysian” posterior probability? I disagree that 0.866 can be called well supported. Usually, Bayesian posterior probability above 0.95 are considered strong and an average of 0.866 could at best be called “with some support”.

Line 262 So cool that this is such a clear pattern, especially for the Plio-/Pleistocene! However, the data presented here is only derive from *Saxifraga* and I think implying that this can nevertheless be applied to “the [entire] alpine flora” is overselling the results. While probably true a more careful wording should be considered.

Line 272 I see no evidence presented by the authors for instability as a cause for upslope biome shifts. Please remove from statement.

Line 278 “studies in other groups, [...] can now be undertaken” that is very gracious of the authors,

but I would recommend rephrasing to reflect that additional data would be required to test the generality of the proposed "Saxifraga model" presented here.

Line 279 I believe that it is impossible to present a general framework for alpine plant evolution based on a single genus and recommend to not oversell the results. The study is a shining example of systematics and biogeography and does not be lifted above its potential.

Line 310 Why does this need to be assumed? This is the first time that topography is named as a potential significant contributor and I would recommend including this either earlier as well or remove it at this point.

Line 315 Citations? I feel that a concept like "species-pump effect" should be introduced earlier than the end of the discussion. Maybe consider removing altogether? If kept and appropriately introduced a discussion again the results of the study would improve the quality of this statement.

Line 319 I can't see that a significant decrease in species rate is present when looking at 2b, this is furthermore not discussed in the results and therefore oddly placed here. Please show the significance and include this as a result maybe in line 212.

Line 320 "established idea of rapid evolutionary diversification in alpine habitats" Maybe this is not totally reflecting accurately the authors. I see the fault at Hughes and Atchison as they misrepresent alpine in the title but write about mountain radiations. I would recommend ignoring that work or more carefully reflect that they do not really present alpine but high mountain data/discussion! I totally comment the author of this manuscript to have a greater focus on environments above the tree line!

Line 322 I totally love Saxifraga and disagree that Lupinus is the most iconic plant radiation in a mountain system!

Line 337 "decoupling between adaptation and speciation". I feel that the authors are rambling on a bit here. Please restrict your discussion to the data at hand.

Line 342 Sure, all sort of further studies can build on this, but I wonder if it is helpful to write this here?

Line 343 "The importance of integration across spatial scales" I'm not sure I understand what the authors would like to say here. Please rephrase.

Line 344 I would like to recommend using the rich literature on long distance dispersal islands here and also recommend going beyond only looking at the Qinghai-Tibet Plateau.

Line 345 "investigate the effect of different ecological scales" again I am a bit confused as what the authors would like to say here. Please rephrase. It feel to me that the authors are trying to make a point at the very end that is not properly introduced in the introduction nor methods or results and maybe a remnant of an earlier version of the manuscript?

Line 347 what do the author mean by characteristics?

Line 349 As above I would recommend against using "defining" in environmental factors typical for alpine environments as the driving factor seems to be average temperatures. I guess the authors are trying to embark on the discussion of factors that are limiting the ability of plants to grow in alpine environments? I would recommend to properly build on this part of the discussion but would need a more structured discussion about the role of adaptive trades and would need a red line from the introduction through to the discussion. This might be difficult to archive without getting distracted from the main findings of the study and might be better suited as the topic of a separate study.

Line 353 I dislike overselling so "much of the World's alpine floras" is difficult for me to agree when presented with results from a single genus.

Line 354 why not use evolutionary terms or plant biogeographic terms?

Line 354 the whole genus is late-Cenozoic isn't it that the surge came in the Plio-/Pleistocene?

Line 356 I don't see the irony in this, please rephrase.

Line 361 Please include a citation for the shrinking alpine habitats.

Line 361 why "on balance" please rephrase.

Line 366 Please include a line on outgroup choice or was there enough overlap between Folk et al. loci and ours?

Line 453 Are there simply no species occurring in only one area? Or what did you do with widespread species?

Line 466 I would find it extremely useful if some of the issues discussed could be summarized here. As pointed out the definition of alpine is not always stringently used especially in areas where mountain grassland habitats exist which are treeless but nevertheless not above the treeline. I totally support a narrower definition of alpine.

Supplement material: I seem to have overlooked the figure legends for the supplement figures and are unable to assess these.

It is reassuring to see a good success rate DNA extraction from herbarium vouchers leading to the much-needed coverage of species from the species tree!

Reviewer 1:

*This study, submitted by Carruthers et al., is an attempt to understand the origins of alpine plant diversity and their changing dynamics through time using the widespread alpine genus *Saxifraga* as an example. Specifically, this study examined the upslope biome shifts, mountain hopping, and alpine speciation processes on alpine plant diversity. The results found that upslope biome shifts are more common than mountain hopping throughout the evolutionary history of *Saxifraga*. Meanwhile, this study found that alpine specialists have lower speciation rates than generalists. I really like this study and am very interested in the scientific questions and study system presented in the current study. However, I have several concerns about the method part.*

Comment: *First, how the current biogeographical framework is suitable for answering the proposed scientific questions on alpine biome assembly needs to be more specific. Here, the author actually modeled the speciation, colonization, and upslope processes among regions for alpine lineages of *Saxifraga* but not the process among real mountain systems (as shown in Figure 1a). For example, Asia includes subtropical mountain regions to cool temperate regions and even islands. This is definitely not a single mountain system, both presently and geologically. The definition of regions may bias the present results. This needs more explanation and details.*

Response: We have removed usage of the phrase mountain system when referring to results from our model. The only remaining usage occurs in parts of the introduction when referring to mountain systems in general (e.g. **lines 65, 68, 119, 122, 126**). We have also altered Figure 1a so that it now does not conflate a single mountain system with a region and thus reflects the details of our model. When referring to the analyses we undertake we refer explicitly to inter-regional mountain hopping (e.g. **lines 152 onwards**).

Comment: *Second, this study found a lower speciation rate of alpine lineages and upslope biome shift are more prevalent than dispersal between regions, which is partially contrary to a previous study by Ding et al. (2020). Ding et al. (2020) found that within the Tibet-Himalaya and Hengduan region (THH), alpine lineages in Hengduan Mountain and Himalaya have a higher in situ speciation than local recruitment (similar to upslope biome shift in this study) and colonization (similar to mountain hopping in this study). While the assembly of alpine lineages in the central Qinghai-Tibet plateau is dominated by colonization processes mainly from Hengduan Mountain. Their results suggested that the origins of alpine plant diversity and their changing dynamics through time could be different in different mountain systems. In the current study, the author classified 5 biogeographical regions but did not distinguish the potential differences of the underlying processes among these regions. For example, Asia and Europe contain over 90% of total species of *Saxifraga*. And Asia contains more alpine species but Europe contains more non-alpine species. The underlying process for the assembly of alpine plants of *Saxifraga* could be different. For Arctic America Caucasus, which contains fewer species. I suggested to highlight more on the differences between the present study and Ding et al. (2020) and also add more details. For example, the present study used the genomic data while Ding et al (2020) used sequences of small DNA fragments that were reported by other groups. In addition, the species distributions are and the defined biomes of two studies could be compared.*

Response: We discuss in more detail the differences between our study and previous studies, and differences between different regions at **lines 346-357**, and **lines 373-414**. The sections from **lines 373-414** have undergone substantial rewriting.

Comment: *Third, this study only focuses on one taxonomic group; therefore, the conclusion is probably taxonomically specific. This caveat could be mentioned in*

Response: We more clearly describe the clade specificity of our findings, and describe the benefits of analyses of other clades at **lines 332-336**, and also to some degree with our more in-depth discussion at **lines 372-397**.

Other minor comments: There are no figure indexes and legends for supplementary figures.

Reviewer 2:

*The study investigates patterns in alpine flora assembly using a time-calibrated phylogeny of *Saxifraga*, a flowering plant clade distributed almost globally in mountain systems. Three potential processes are investigated, upslope biome shifts, mountain hopping, and in situ alpine speciation. The relative importance of these processes is analysed by categorizing species into five geographic regions and two biome categories (alpine and non-alpine), and by comparing 'specialists' (alpine or non-alpine) and 'generalists' (species occurring in both biome categories). Estimated are (1) the rates of shifts among biome categories and (2) speciation rates per biome category. Shift rates are compared for 'specialists' and 'generalists' in total and in four time categories (5, 5–10, 10–15, and >15 Ma). Results indicate that (i) upslope biome shifts are ~3 times higher than that of mountain hopping and that these shift frequencies are accelerated in the last 5 Ma, and (ii) generalists show higher speciation rates than specialists (low 'alpine speciation'). Additionally, lineage-specific speciation rates are estimated, and results indicate that (iii) periods of speciation rate acceleration (at ca. 30 and 8 Ma) in *Saxifraga* predate biome shift acceleration (≤ 5 Ma). The results are discussed in the context of Late-Cenozoic climate cooling, which likely correlates to a spatial extension of alpine areas, as well as Pliocene climate fluctuations. Both are discussed as drivers of upslope biome shifts and these shifts are suggested as the process that primarily generates alpine plant diversity. From the results presented it seems valid to conclude that "alpine floras of individual mountain systems are primarily derived from non-alpine floras via upslope biome shifts" (line262f).*

Comment: *A flaw here is that the authors do not explicitly state that "mountain hopping" is analysed only between the five global geographic regions and not within "mountain systems," which could, or certainly would, increase the rate of alpine-to-alpine mountain dispersals. This makes the 'mountain hopping' results only valid for global geographic scales, which should be mentioned.*

Response: As discussed in response to reviewer 1 we have significantly clarified the manuscript with respect to this issue (removing mentions of mountain system where it is unnecessary or misleading, changing Figure 1a, and referring to inter-regional mountain hopping when referring to our analyses). Whilst we recognise that altering the scale of our analysis (or any analysis of this nature) could affect the results (increasing the rate of mountain hopping for example) we have added the point that our analytical framework is actually biased toward mountain hopping at **lines 321-323**. This adds additional weight to the key conclusions from our study.

Comment: *What is less clear, however, is whether this "upslope biome shift" (mainly montane to alpine?) will at all entail ecological adaptation. The alpine biome is defined in reference to trees (that do not grow there) and therefore has ecological implications primarily for trees. Whether this holds also for herbs is not investigated in the study and seems at least questionable. For example, a rocky cliff habitat some hundred meters below the tree line might have quite similar abiotic conditions to a similar habitat in the alpine zone (see e.g., Körner & Hiltbrunner 2021 Diversity 13, 383). With this in mind, the statement "that a decoupling between adaptation and speciation may be prevalent in plant macroevolution" (line 337) is not supported by the evidence presented.*

Response: We assert that shifts into the alpine zone are likely to be adaptively significant (regardless of the existence of these other "similar" areas) on the basis that alpine zone specialists are, by definition excluded from these areas. Some aspect of specialisation to the alpine zone appears to exclude these lineages from these areas (**lines 411-414**).

Comment: What is less clearly presented is the “Biogeographical reconstruction” (SI line 19) that relies on the ClaSSE model illustrated in Figure S8. A short paragraph (in the SI?) summarizing how mountain hopping and upslope biome shifts are actually obtained from the ClaSSE estimates would be appropriate (also, how was the model set up, e.g., from what kind of distribution did you draw transition rates and diversification rates? et cetera).

Response: We have added additional methodological detail at **lines 540-546**. Links to Zenodo and Github are **lines 563-564**.

Comment: Also, while assessing submitted data, I was slightly surprised that no time-calibrated tree (the ‘key’ framework), no ‘easy-to-access’ data set of regional and biome preferences, and no analytical scripts (RevBayes, etc.) are provided. This ‘missing’ data does not seem essential for my report but would be great for the scientific community if provided upon publication.

Response: **Lines 563-564**. Biome and regional preferences are provided in supplementary table S5.

Comment: Unluckily, both approaches to account for branch length uncertainty do explicitly ignore the topological information that is in conflict with the inferred species tree. It is unclear to me how phylogenetic uncertainty when fully accounted for would influence results. Uncertainty in divergence time estimates is, as much as possible, convincingly accounted for (see Figs S3, S4, S6, S7). But how does topological uncertainty influence estimated shift rates is neither investigated (certainly, there are computational limits) nor discussed. A statement discussing this potential source of rate estimate biases should be appropriate.

Response: We have added a paragraph to the results section where we describe how topological uncertainty in the species tree does not increase significantly toward the present (**lines 267-276**). There is also a new supplementary figure to illustrate this – Figure S5.

The ClaSSE model is a valid approach and accounts for unsampled or extinct diversity in analyses of biogeographic range evolution.

The manuscript presents very clearly structured arguments in the introduction and discussion and the reasoning is easy to follow throughout. Besides, sometimes a little more understatement would strengthen the narrative and provide more confidence in the interpretation of results and the conclusions (see, e.g., second paragraph of this review).

Appropriate literature is referred to in the manuscript.

Check ms throughout (also figures) for consistent usage of Ma/Myr.

Hope these comments are helpful. Very much looking forward to seeing this study published!

Reviewer 3:

General comments: This is a rare study with a high sampling percentage and in general a well conducted study of an amazing group that deserves to get published in Nature Communications!

Comment: *The text could improve by some careful rewording. I included some line-by-line comments. The term alpine and high elevation environments are unfortunately not always used very stringently in literature, and it is easy to compare apples with pears. I would recommend setting this dilemma out in the introduction and read the published literature carefully regarding the definition used and apply it stringently.*

Response: As in response to other reviewers, the precision with which we describe the study system throughout the introduction is significantly increased.

Comment: I appreciate that the authors seem to have worked extensively on the Qinghai-Tibet Plateau however a more balanced approach to all regions in which *Saxifraga* occurs might be useful when trying to present this as a world-wide model of alpine evolution.

Response: Referencing throughout the text, and especially in the introduction, has been broadened considerably. As such there are many more examples from other regions where appropriate. This exercise was especially useful for better placing our study in its broader context. Examples of such changes are at **lines 97-103, 131-136, 347-349, 403-405.**

Comment: Line 78. this paragraph described well studied processes that do not exclusively occur in alpine habitats but are poorly referenced here with an unfitting citation of a paper looking at orogeny and climatic influence in Tibet-Himalaya-Hengduan mountains. Please reconsider more appropriate citations of the rich scientific literature on the topic!

Response: Additional and relevant references added at **lines 75 and 86.**

Comment: Line 86. "colonizing" is a very distinctive term in plant biology in addition to social connotation that might be unwanted. Colonizing is probably best reserved for the process of establishing in a previously uninhabited area. As such "colonizing" would not be wrong but I would recommend using immigration instead.

Response: Rephrased. Now at **line 83.**

Comment: Line 89. while indeed "The relative importance of these processes is challenging to unravel and highly context- and scale-dependent." There is work available that should be at least cited here such as Gehrke 2018 Staying cool: Preadaptation to temperate climates required for colonising tropical alpine-like environments.

Response: Reference added at **line 86.**

Comment: Line 91. I would recommend looking at the literature on tree line more closely as the main factor is not "temperature fluctuations, intense ultraviolet radiation, short growing seasons, and low nutrient soils". Treeline seem to be mainly driven by average temperature and especially average soil temperature! Körner & Paulsen 2004 A world-wide study of high-altitude treeline temperatures. <https://doi.org/10.1111/j.1365-2699.2003.01043.x> And I would recommend including this fact as not to confuse characterised with causing in reader unfamiliar with the subject. I believe the authors wish to express here that plants growing in alpine environments often experience "temperature fluctuations, intense ultraviolet radiation, short growing seasons, and low nutrient soils".

Response: I agree that the way we had phrased it in the earlier version was misleading, and the reviewer is correct to point out that we referring to what the alpine lineages are exposed to. We have therefore amended the text at **lines 88-90.**

Comment: Line 105. Both citations point toward work done on Tibetan plant lineages, I find it difficult to accept that as relevant enough to make such a broad general statement here. Please add relevant citations from other parts of the world or remove. This is especially true as the cited work from Favre does not prove that dispersal happened from alpine environments. The publication focuses on geography not climate reconstructions and includes both alpine and subalpine as a single biome.

Response: This section is significantly re-written with additional references at **lines 87-103.**

Comment: Line 107 I would recommend to rewrite this as the sentence is confusing ("implies mountain hopping may also be infrequent"?) and there are surely more elegant ways to present this.

Response: Rephrased, **lines 106-107.**

Comment: Line 110. I find work on island biogeography much more relevant here. There are some beautiful studies such as Alsos et al. 2007 Frequent Long-Distance Plant Colonization in the Changing

Arctic. DOI: 10.1126/science.1139178 - not really a macroevolutionary study, or anything to do with mountains.

Response: I haven't specifically used this citation, but with the broader sample of citations now used in the introduction, I think the citations here are sufficient (**lines 105-116**).

Comment: Line 112. I would move citation 19 to before the full stop and instead add Gehrke & Linder 2014 Species richness, endemism and species composition in the tropical Afroalpine flora.

Response: The citations currently used are most relevant to the patterns being discussed.

Comment: Line 126. It might be useful to incorporate here the results of Brochmann, et al. History and evolution of the afroalpine flora: in the footsteps of Olov Hedberg. *Alp Botany* 132, 65–87 (2022).

Response: Reference added at **line 126**.

Comment: Line 134. Not all of the tree free Andean habitats are alpine habitats, and I would recommend not to use citation 12 here as that paper does not proof this point. I would even doubt that 27 is really proving "alpine speciation rates being elevated relative to background rates". However, this is not easy and could also be addressed by more careful wording.

Response: This section (now **lines 118-136**) is substantially updated with more appropriate citations.

Comment: Line 145. Please add citation.

Response: Citation added at **line 146**.

Comment: Line 149: "mountain hopping" sound repetitive and directional maybe long-distance dispersal might be better used? No matter what the authors decide a definition would be great, it this only between larger geographical regions or how is long-distance dispersal/mountain hopping identified? Fig.2 would indicate that the authors only considered changes on continental scale?

Response: The phrase is defined extensively in the introduction, even more clearly now than it was previously. Also, the illustration of the concept in figure 1 is now clearer.

Comment: Line 153. I'm not a native English speaker but precipitate has a negative, foreboding connotation for me. You might want to check this.

Response: Changed. Now **lines 154-155**.

Comment: Line 158 It is a bit unusual to have a paragraph with conclusions at the end of the introduction, but I leave that one to journal style guides.

Response: I agree, but it is often the style with which articles are written in the journal.

Comment: Line 161 pressure or maybe opportunity (well my glass I half full 😊)

Response: Changed. Now at **line 162**.

Comment: Line 163 see comment on line 153 but since the authors see pressure rather than opportunity maybe increased species rate is indeed negative?

Response: Sentence removed.

Comment: Figure 1: Beautiful! It might be ignorant to new changes but is *Saxifraga hederifolia* from Ethiopia no longer accepted?

Response: African species not included in the study system. Added in methods at **lines 505-507**.

Comment: Line 190 "Baysian" posterior probability? I disagree that 0.866 can be called well supported. Usually, Baysian posterior probability above 0.95 are considered strong and an average of 0.866 could at best be called "with some support".

Response: Rephrased. **Lines 182-184.** And further discussion of tree topology support at **lines 267-276.**

Comment: *Line 262 So cool that this is such a clear pattern, especially for the Plio-/Pleistocene! However, the data presented here is only derive from Saxifraga and I think implying that this can nevertheless be applied to “the [entire] alpine flora” is overselling the results. While probably true a more careful wording should be considered.*

Response: We have rephrased and toned down the beginning of our discussion a little.

Comment: *Line 272 I see no evidence presented by the authors for instability as a cause for upslope biome shifts. Please remove from statement.*

Response: We have rephrased to refer more clearly to climatic instability (now **line 361**).

Comment: *Line 278 “studies in other groups, [...] can now be undertaken” that is very gracious of the authors, but I would recommend rephrasing to reflect that additional data would be required to test the generality of the proposed “Saxifraga model” presented here.*

Response: Entirely rephrased at **lines 334-336.**

Comment: *Line 279 I believe that it is impossible to present a general framework for alpine plant evolution based on a single genus and recommend to not oversell the results. The study is a shining example of systematics and biogeography and does not be lifted above its potential.*

Response: As above, rephrased.

Comment: *Line 310 Why does this need to be assumed? This is the first time that topography is named as a potential significant contributed and I would recommend including this either earlier as well or remove it at this point.*

Response: Removed.

Comment: *Line 315 Citations? I feel that a concept like “species-pump effect” should be introduced earlier than the end of the discussion. Maybe consider removing altogether? If kept and appropriately introduced a discussion again the results of the study would improve the quality of this statement.*

Response: This is probably now clearer with the more careful introduction of climatic instability earlier in the discussion. Now **lines 370-371.**

Comment: *Line 319 I can’t see that a significant decrease in species rate is present when looking at 2b, this is furthermore not discussed in the results and therefore oddly placed here. Please show the significance and include this as a result maybe in line 212.*

Response: Language clarified at **lines 374-375.**

Comment: *Line 320 “established idea of rapid evolutionary diversification in alpine habitats” Maybe this is not totally reflecting accurately the authors. I see the fault at Hughes and Atchison as they misrepresent alpine in the title but write about mountain radiations. I would recommend ignoring that work or more carefully reflect that they do not really present alpine but high mountain data/discussion! I totally comment the author of this manuscript to have a greater focus on environments above the tree line!*

Response: Clarified at **line 376-377.**

Comment: *Line 322 I totally love Saxifraga and disagree that Lupinus is the most iconic plant radiation in a mountain system!*

Response: removed.

Comment: Line 337 “decoupling between adaptation and speciation”. I feel that the authors are rambling on a bit here. Please restrict your discussion to the data at hand.

Response: We hope the greater focus that now exists in the discussion demonstrates the relevance of this statement more clearly. Now rephrased at **lines 391-397**.

Comment: Line 342 Sure, all sort of further studies can build on this, but I wonder if it is helpful to write this here?

Response: This paragraph is significantly rephrased with greater clarity (now **lines 399-414**).

Comment: Line 343 “The importance of integration across spatial scales” I’m not sure I understand what the authors would like to say here. Please rephrase.

Response: See response to comment above.

Comment: Line 344 I would like to recommend using the rich literature on long distance dispersal islands here and also recommend going beyond only looking at the Qinghai-Tibet Plateau.

Response: We have added significant additional literature from other regions.

Comment: Line 345 “investigate the effect of different ecological scales” again I am a bit confused as what the authors would like to say here. Please rephrase. It feel to me that the authors are trying a make a point at the very end that is not properly introduced in the introduction nor methods or results and maybe a remnant of an earlier version of the manuscript?

Response: This paragraph is now significantly rephrased.

Comment: Line 347 what do the author mean by characteristics?

Response: Removed.

Comment: Line 349 As above I would recommend against using “defining” in environmental factors typical for alpine environments as the driving factor seems to be average temperatures. I guess the authors are trying to embark on the discussion of factors that are limiting the ability of plants to grow in alpine environments? I would recommend to properly build on this part of the discussion but would need a more structured discussion about the role of adaptive trades and would need a red line from the introduction through to the discussion. This might be difficult to archive without getting distracted from the main findings of the study and might be better suited as the topic of a separate study.

Response: Rephrased.

Comment: Line 353 I dislike overselling so “much of the World’s alpine floras” is difficult for me to agree when presented with results from a single genus

Response: Rephrased, now at **lines 416-418**.

Comment: Line 354 why not use evolutionary terms or plant biogeographic terms?

Response: Rephrased, **line 418**.

Comment: Line 354 the whole genus is late-Cenozoic isn’t it that the surge came in the Plio-/Pleistocene?

Response: Changed, **line 418**.

Comment: Line 356 I don’t see the irony in this, please rephrase.

Response: Neither do I, removed.

Comment: Line 361 Please include a citation for the shrinking alpine habitats.

Response: Added at line 424.

Comment: Line 361 why “on balance” please rephrase.

Response: Changed at line 424.

Comment: Line 366 Please include a line on outgroup choice or was there enough overlap between Folk et al. loci and ours?

Response: Added line 433-435.

Comment: Line 453 Are there simply no species occurring in only one area? Or what did you do with widespread species?

Response: It says specimens, not species, a specimen can only occur in one region

Comment: Line 466 I would find it extremely useful if some of the issues discussed could be summarized here. As pointed out the definition of alpine is not always stringently used especially in areas where mountain grassland habitats exist which are treeless but nevertheless not above the treeline. I totally support a narrower definition of alpine.

Response: We feel the latest version of the manuscript sufficiently defines the alpine zone.

Comment: Supplement material: I seem to have overlooked the figure legends for the supplement figures and are unable to assess these.

Response: I don't know why these were lost from the first submission. I have pasted them onto the end of both the supplementary document and the main manuscript.

It is reassuring to see a good success rate DNA extraction from herbarium vouchers leading to the much-needed coverage of species from the species tree!

Reviewers' Comments:

Reviewer #1:

Remarks to the Author:

I have no further concern.

Reviewer #2:

Remarks to the Author:

All comments were given due consideration